



# A Repository of 100+ Years of Measured Soil Freezing Characteristic Curves

Élise Devoie[1,2], Stephan Gruber[2], and Jeffrey M. McKenzie[1]

[1]McGill University, Earth and Planetary Sciences, Montréal, Canada
[2]Carleton University, Geography and Environmental Studies, Ottawa, Canada

**Correspondence:** Élise Devoie (elise.devoie@mcgill.ca)

**Abstract.** Soil freeze-thaw processes play a fundamental role in the hydrology, geomorphology, ecology, thermodynamics and soil chemistry of a cold-regions landscape. In understanding these processes, the temperature of the soil is used as a proxy to represent the soil ice content through a soil freezing characteristic curve (SFCC). This mathematical construct relates the soil ice content to a specific temperature for a particular soil. SFCCs depend on many factors including soil properties (e.g., porosity, composition, etc.), soil pore water pressure, dissolved salts, (hysteresis in) freezing/thawing point depression, and degree of saturation, all of which can be site-specific and time varying. SFCCs have been measured using various methods for diverse soils since 1921, and to date this data has not been broadly compared, in part because it has not previously been compiled in a single data set. The dataset presented in this publication includes SFCC data digitized or received from authors, and includes both historic and modern studies. The data is stored in an open-source repository, and an R package is available to facilitate its use. Aggregating the data has pointed out some data gaps, namely that there are few studies of coarse soils, and comparably few *in-situ* measurements of SFCCs in mountainous environments. It is hoped that this dataset will aid in the development of SFCC theory, and improve SFCC approximations in soil freeze-thaw modelling activities.

## 1 Introduction

Soil freeze-thaw processes play a fundamental role in the hydrology, geomorphology, ecology, thermodynamics and soil chemistry of cold-regions landscapes. Due to a changing climate, changes in the timing and magnitude of freeze-thaw events are altering cold-regions systems (Meredith et al., 2019). Modelling tools are needed to predict, understand, and develop adaptive strategies to these changes. Many models of soil freeze-thaw exist (e.g. CAst3M, SUTRA-ICE, SMOKER, FEEFLOW-FTM, and GeoSlope to name a few), but all physically-based models including partially frozen soils rely on a Soil Freezing Characteristic Curve (SFCC) to describe the relationship between the unfrozen water content and the temperature in freezing soils (Rühaak et al., 2015; Grenier et al., 2018; McKenzie et al., 2007; Anbergen et al., 2015).



## 2 Background

### 2.1 Understanding an SFCC

An example SFCC is shown in Figure 1. The unfrozen water content is reported on the y-axis, measured in units of m³ unfrozen water per m³ of soil, but equally valid as a mass fraction or saturation. Theoretically, the unfrozen water content is a smooth,

25 monotonic function of temperature. There are three key features in an SFCC: the initial water content, the bulk melting point, and the residual water content. The total water content is the liquid water in the sample when it is completely thawed and encompasses the pore water and water bound to soil particles, independent of solute presence. The bulk melting temperature is the temperature at which phase change is complete when thawing the soil. This temperature is generally depressed below the melting point of pure water, $0°$ C, due to pore geometry and solutes. Many studies report a freezing point depression, which

30 describes the initiation of phase change on the freezing limb of the SFCC, generally colder than the bulk melting temperature (Gharedaghloo et al., 2020; Liu and C. Si, 2011). This hysteresis persists through the SFCC as pore shape affects ice nucleation, as is widely reported in SFCC literature (Kruse et al., 2018b; Chai et al., 2018; Tian et al., 2014; Koopmans and Miller, 1966; Kozlowski, 2009; Hu et al., 2020). Finally, the residual water content is the 'unfreezable' water content in the soil; water that is so tightly bound to the soil particles that it remains energetically favourable to remain in liquid form even at temperatures

35 tens of degrees below zero (Wettlaufer and Worster, 2006). These features define the shape of the SFCC, and vary between soil type, saturation, measurement technique and sample preparation.

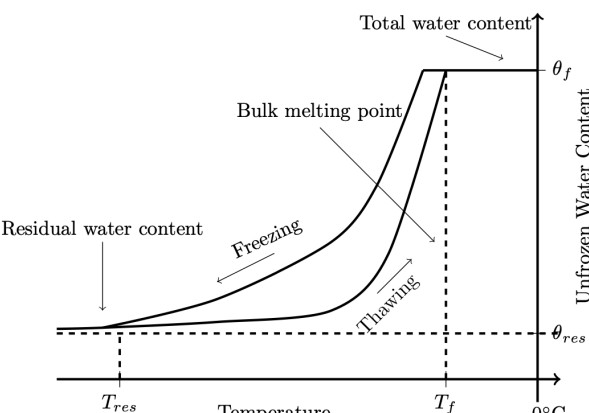

**Figure 1.** A typical soil freezing characteristic curve, composed of separate freezing and thawing limbs to form a hysteretic loop. Steepness, shape and residual water content depend on soil properties.





## 2.2 Constructing an SFCC

The SFCC parameterizes the effects of several physical processes that control the unfrozen water content of soils. Along the freezing limb of the SFCC (top curve in figure 1), the first water to change from liquid to solid is the pore water. Neglecting solute content and considering pore size alone, this water freezes according to the Gibbs-Thompson Equation:

$$T_m = T_0 - \frac{\gamma_{sl} T_0}{\rho_i L_f} \frac{2}{R'} \tag{1}$$

where $T_m$ is the melting point temperature of water in soil (K), $T_0$ is the melting point temperature of pure liquid water (273.15 K), $\gamma_{sl}$ is the free energy coefficient of the ice-water interface (0.029 J/m$^2$), $\rho_i$ is the ice phase density (917 kg/m$^3$), $L_f$ is the latent heat of phase transformation ($3.35 \times 10^5$ J/kg), and $R'$ is the pore radius (m). The Gibbs-Thompson Equation implies that water in large pores freezes first (Teng et al., 2021; Sliwinska-Bartkowiak et al., 1999; Koopmans and Miller, 1966). This effect contributes to the steepest part of the curve shown in figure 1 (Tian et al., 2014). As the curve flattens, the unfrozen water content is composed progressively of more adsorbed water (Tian et al., 2014) that is tightly bound to the soil particles, and depending on the soil type, may form a film that completely separates the ice in the pore space from the soil grains (Jin et al., 2020). Soils in which this liquid water film occurs are termed 'solid-liquid-solid' (SLS) soils and are predominantly clays, where soil particles often carry a negative electrical charge (Koopmans and Miller, 1966). Alternatively, 'solid-to-solid' (SS) soils have pore ice in contact with soil particles, such as occurs in sandy soils (Koopmans and Miller, 1966). The soil type (SLS or SS) defines the residual water content - the liquid water left that does not freeze (or freezes extremely slowly) - as temperature continues to decrease (Jin et al., 2020). For example, studies of clays have found substantial liquid water fractions on the order of 20 m$^3$/m$^3$ at temperatures as low as -25°C (Bittelli et al., 2004b; Kong et al., 2022).

There is a wide range of SFCCs as a result of the complex relationships between soil texture, pore geometry, solute content, freezing point depression, soil saturation and other physical properties of freezing soils. These physical factors make SFCCs difficult to predict accurately using empirical models that are limited to specific reference or measured soils (Mu et al., 2018; Amiri et al., 2018). Physical models are intended to be more transferable, but rely on detailed soil parameters such as the specific surface area of the soil, grain size analysis, polarity of soil particles and other hard-to-collect data (Wang et al., 2017; Bai et al., 2018). It is impractical to obtain this type of distributed information for soils which are generally heterogeneous and spatially variable. Though some research has improved the fitting of models to SFCCs (e.g. Hu et al.), the SFCC data for comparison is generally limited, and no compilation exists reporting SFCC data such that it might be compared in similar units based on common metadata information.

## 2.3 Objective

The aim of this article is to present a repository of SFCC data found in the literature. This repository is aimed to improve estimates of SFCCs for use in cold regions modelling applications and to provide potential SFCCs for sites where local data is not available. We provide a compilation of the data, including the relevant metadata, alongside an R package for convenient data extraction and processing.



## 3 Data Aggregation

Literature reporting SFCCs from 1921 to August 2021 is reviewed, and measured SFCCs are either digitized from the figures in publications using the digitizer tool in OriginPro 2021 (Research Lab), or the data is requested from the authors. In total, 254 publications with relevant topics are identified, and from these, 416 SFCCs are extracted. For each curve, metadata is also captured, including soil data, measurement technique, and experimental conditions, when available. The digitized curves along with their metadata are stored in an open-source archive for future use, and an R package containing the data and data tools is

discussed in section 5.

## 4 Metadata

The metadata captured alongside the SFCCs identifies the measurement procedure (e.g. measurement technique, sample preparation) and the characteristics of the sample to be measured. These characteristics include soil textural data, specific surface area, porosity, and density, when available. The metadata allows potential users to extract SFCC data relevant to a particular

soil to estimate the SFCC more accurately. Additional metadata describing measurements made on the warming (thawing) or cooling (freezing) of samples, the units of the original measurement, and the degree of saturation of the sample are included to improve our understanding of the physical freeze-thaw process, and to better interpret the data reported. In the following sub-sections, each metadata field is described.

### 4.1 Measurement Methods

Many methods are used to measure SFCCs. The most common methods use the electrical properties of the soil to infer the liquid and frozen water content (Capacitance, Resistivity, Permitivity) as they are fairly easy to use in laboratory and field settings (Bittelli et al., 2004b; Romanovsky and Osterkamp, 2000; Fabbri et al., 2006; Wu et al., 2017; Hu et al., 2020). Some methods focus on measuring the unfrozen water content based on density or physical properties of the soil including dillatometry and ultrasound (Koopmans and Miller, 1966; yan Wang et al., 2006). Yet others rely on the atomic structure and energy of the

water molecules including NMR, gamma spectroscopy, XRD and Neutron Scattering (Watanabe and Wake, 2009a; Jame and Norum, 1980; Christenson, 2001). These atomic-level methods can distinguish between the types of unfrozen water content as tightly bound water has different polarization (Watanabe and Wake, 2009a; Chen et al., 2021). Finally the thermal properties of the soil water can be used to establish the unfrozen portion using thermal conductivity, calorimetry (isothermal or idiabatic), Differential Thermal Analysis (DTA) or Differential Scanning Calorimetry (DSC) (Anderson and Tice, 1972; Kozlowski and

Nartowska, 2013b). The different measurement techniques are classified according to the physical quantity they measure in Figure 2, where it is apparent that most recent measurements are based on the electrical properties of freezing soils, while older methods rely on measurements of the thermal or physical properties of the soil. Table A1 in appendix A presents all of the measurement techniques found in literature, their key characteristics, and the publications which used this method to

Earth System
Science
Data

determine the SFCC. Readers are directed to an accompanying manuscript, Devoie et al. (2022b), for a detailed description of
each measurement technique and the associated error.

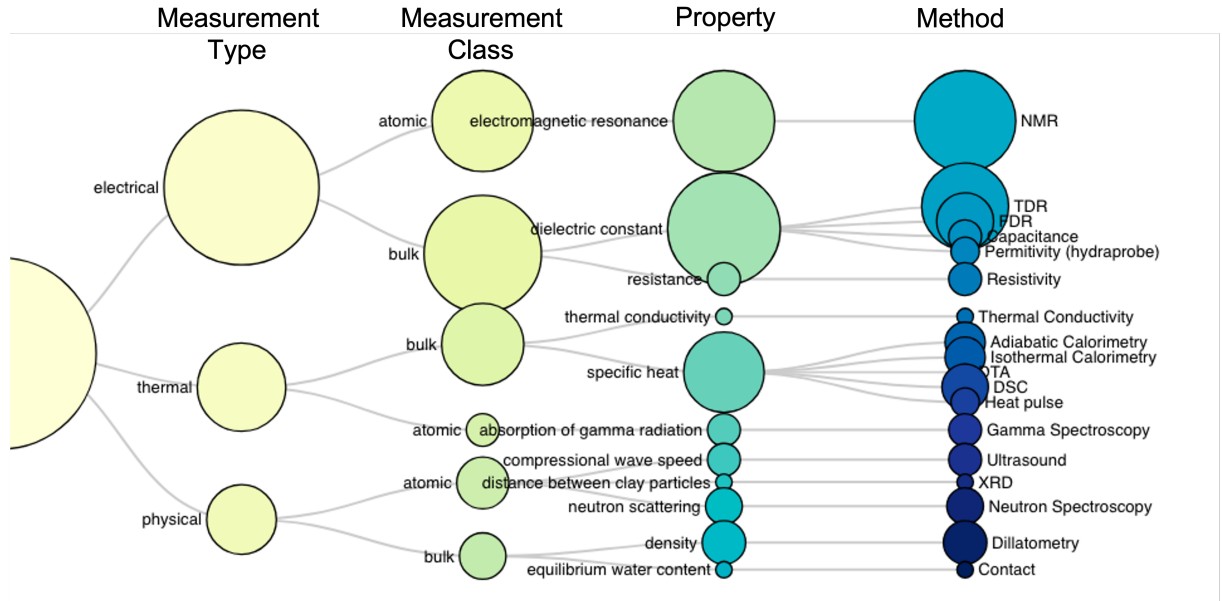

**Figure 2.** Organization of measurement methods according to measurement principle. Size of nodes represents the relative number of SFCCs
collected using each method.

## 4.2   Sample preparation

Though it is possible to use some measurement techniques to measure SFCCs *in-situ*, over 75% of reported SFCCs were
measured in a lab setting. As noted by Buehrer and Rose (1943), when a thawed soil is packed in the lab (reconstituted), its
response to freeze-thaw cycling is dependant on the number of freeze-thaw cycles it has undergone. Kozlowski and Nartowska
(2013a) found that after 5 cycles, the freeze-thaw behaviour of the soils was not yet stabilized. This is a major concern when
applying lab-measured results to modelling field scenarios, and should be considered when measuring and using measured
SFCCs if the lab measurements are performed on a sample that has not undergone prior freeze-thaw cycling.

   Aside from thermal cycling, repacked samples in the lab are often sieved, homogenized, compacted to known densities,
and desalinated, making the results difficult to compare to field measurements. They do not include the effects of soil het-
erogeneity in the small sample volumes tested (Veraart et al., 2016), and represent simplified systems as compared with the
processes which occur in the field. Many studies use silica beads of uniform size and shape with distilled water to simplify
the measurement process, leading to results that are not easily transferable to field conditions. The presented data repository
includes information on the lab or field measurement site, and whether the soil sample has been disturbed (re-packed) before
measurement.



### 4.3 Soil Properties

Most measurements of SFCCs in the literature report some soil characteristics - the name of the soil, its grain size distribution, and/or a textural description. Though these metrics are important, they are not systematically reported, and is difficult to compare between studies. For the purpose of the data repository, data regarding the sand, silt, clay, and organic fractions of the soil is reported when possible. If no quantitative soil textural data was provided, the name of the soil, for example "Walla Walla silt" Bittelli et al. (2004b), is used to classify the soil, in this case as a silt. In this manner, a range of data can be drawn from the repository based on the nature and accuracy of the soil properties required.

Not only is soil texture important in determining the SFCC, but other soil properties play an important role as well. The soil specific surface area (SSA) partially determines the quantity of adsorbed water, which generally correlates to greater residual unfrozen water content (Dillon and Andersland, 1966). The soil porosity determines the maximal water content of the soil. The soil density helps identify and categorize soils when other properties are not reported. The organic fraction is related to the porous structure of the soil, and its capacity to hold water, for example in closed pores present in peat (Rezanezhad et al., 2012). Though few publications report all of these parameters, those that are provided are included in the metadata associated with each entry in the repository.

### 4.4 Hysteresis

Many studies report a difference between the freezing (cooling) and thawing (warming) limb of the SFCC (e.g. Kruse et al. (2018b); Chai et al. (2018); Tian et al. (2014); Koopmans and Miller (1966); Kozlowski (2009); Hu et al. (2020)), but few report data for both the freezing and thawing limb. Hysteresis is soil-dependant and most apparent in fine-grained silt and clay soils (Zhang et al., 2020). The freezing process tends to lag behind the thawing process (as seen in Figure 1) due to ice nucleation which depresses the initiation of freezing (Zhang et al., 2020; Wu et al., 2017), heterogeneity in thermal conductivity and freezing point (Amiri et al., 2018), and the Gibbs-Thomson Equation describing the effect of wetting angle and pore geometry on the freezing point (Gharedaghloo et al., 2020), where highly wetted soils (hydrophylic) have a larger freezing point depression. Ice penetration into small soil pores has higher interface curvature between the liquid and solid phases than melting, and thus occurs at colder temperatures than thawing, contributing to the difference in observed rates (Tian et al., 2014; Chai et al., 2018).

The metadata of each SFCC reports whether the data is collected along the freezing or thawing limb, if included in the original publication. If no data was provided on the direction of phase change, it is most likely that studies in the lab are performed on the freezing limb as this is the case in most lab measurements, and makes up more than half of the data collected, as seen in Figure 3(a).

### 4.5 Saturation

Many measurements of SFCCs are performed on a saturated sample (Figure 3(b)) to avoid the measurement error associated with dry or partially wet samples, for instance the reduced dielectric permitivity in TDR (Watanabe and Wake, 2009a). This


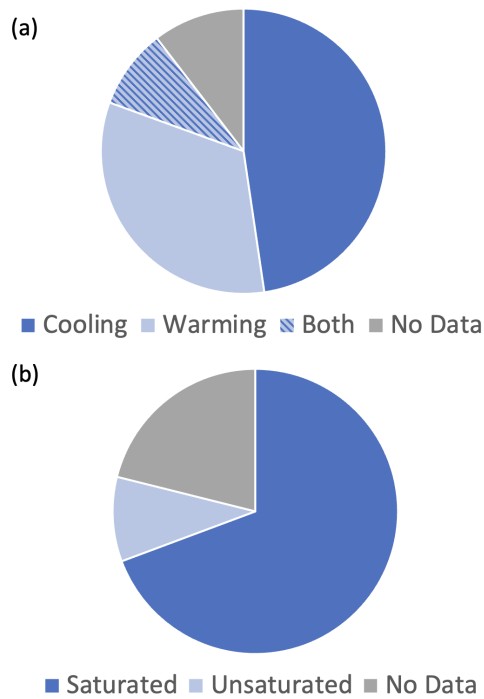

**Figure 3.** (a) Freezing (cooling) or thawing (warming) limb prevalence in SFCC data, (b) initial water content prevalence.

simplifies the measurements as there is only liquid water or ice in the pore spaces. The total saturation of the sample must be presented when reporting an SFCC so that they are comparable. Many authors report the last water to freeze in a soil, i.e., the water that is most tightly bound to the soil particles, is also the last water present in a drying soil (Kurylyk and Watanabe, 2013;

Teng et al., 2020; Zhou et al., 2020; Dall'Amico et al., 2011). This assumption means that saturated SFCCs can be used as an adequate approximation of unsaturated SFCCs given known initial water content, and the same relation between unfrozen water content, pressure (saturation) and temperature can be used for both saturated and unsaturated soils (Dall'Amico et al., 2011).

## 5 Data Tool

An R package is available to facilitate data exploration (Devoie and Brown, 2022). This package contains the repository, as well as some key functions needed to manage the SFCC data. Details on downloading the open-source package, and instructions on how to contribute to the dataset are included in the data availability statement. Contributions to this repository are welcome. This section describes the tools available in this package to manage SFCC data.



## 5.1 Add an SFCC to the repository

The first data tool is a function which takes SFCC data alongside its associated metadata and adds it to a local copy of the existing repository. This file contains the SFCC data and is linked to an index in the metadata file. The metadata file stores all of the information associated with the SFCC. In order to be appropriately interpreted, the SFCC data to be ingested must be prepared according to the data template in Appendix B. The script first verifies that all required data is present and that the data to ingest does not already exist in the archive before adding it. Users are encouraged to contact the authors in order 165 to contribute their datasets to the larger repository, making them available to the wider cryosphere community. This will also ensure the accompanying data tools can be applied to the dataset.

## 5.2 Retrieve an SFCC from the archive

This function returns the SFCC associated with a specified index from the repository. The units associated with this SFCC are temperature in $^\circ$C, and volumetric water content in m$^3$/m$^3$ dry soil. In soils that expand during wetting, e.g., some clays Chai 170 et al. (2018), the volumetric water content may exceed 1.

## 5.3 Find all SFCCs matching criteria

This function takes user-specified search terms, for example: soil textural limits, saturation, specification of field or lab measurement, and others, and combines them to return the indices of the SFCCs that match all (or any) of the specified criteria. This list of indices can then be used with the function described above to return the SFCCs corresponding data.

## 5.4 Error Estimation

The R package is also accompanied by error estimation tools which take all of the information known about the uncertainty in each measurement, including the measurement technique, interactions with various soil types, hysteretic effects, etc.. The error tool provides the main classes of SFCC fitting functions present in the literature and allows the user to obtain uncertainty ranges associated with the data they extract. More detail is provided in the accompanying paper (Devoie et al., 2022b). SFCC 180 type curves, or generalized estimates for common soil textures are also provided.

## 6 Data availability

The SFCC data in support of this publication are openly available as a zenodo archive: https://doi.org/10.5281/zenodo.5592825 (Devoie et al., 2022a), which will be regularly updated as users contribute to the database. The R package, including this data and data tools, is available from the author's github: https://github.com/egdevoie/SFCCRepository (Devoie and Brown, 2022), 185 and work is underway to publish this as a CRAN package.



# 7 Conclusions

The data repository presented includes soil freezing characteristic curves collected from literature dating between 1921 and 2021. This data is presented as a zenodo archive, but can also be accessed alongside metadata describing each SFCC in an open-source repository designed as an R package (Devoie et al., 2022a; Devoie and Brown, 2022). This package facilitates

SFCC data usage, including adding new SFCCs, retrieving SFCC data matching user-specified characteristics, and estimating the error associated with SFCC data. This repository will be useful in modelling soils as it allows to estimate not only SFCCs, but also to identify the uncertainty associated with these estimates.





**Appendix A: SFCC Measurement Techniques**





**Table A1.** Summary of measurements included in data repository organized by property measured and measurement method

| Property Measured | Method | References |
|---|---|---|
| Electromagnetic resonance | Nuclear Magnetic Resonance: measure of the relaxation frequency of hydrogen atoms in water | Akimov (1978); Black and Tice (1989); Chai et al. (2018); Darrow et al. (2009); Dongqing et al. (2016); Ishizaki et al. (1996); Kruse et al. (2018a); Kruse and Darrow (2017); Ma et al. (2017); Nakano et al. (1982); Smith and Tice (1988); Stillman et al. (2010b, a); Suzuki (2004); Teng et al. (2020); Tian et al. (2014); Tice (1978); Tice et al. (1982); TOKORO et al. (2010); Wang et al. (2021b, a); Watanabe and Wake (2009a); Yoshikawa and Overduin (2005); Zhang et al. (1998); Zhou et al. (2020) |
| Dielectric constant | Time Domain Reflectometry: timing of reflected signal dependant on dielectric constant of soil | Azmatch et al. (2012); Christ and Kim (2009); Christ and Park (2009); Flerchinger et al. (2006); Hivon and Sego (1990); Konrad and Duquennoi (1993); Mu et al. (2019); Nagare et al. (2012); Patterson and Smith (1981a, b); Roth and Boike (2001); Smith and Tice (1988); Spaans and Baker (1996); Stähli et al. (1999); Stillman et al. (2010a); Watanabe and Mizoguchi (2002); Watanabe and Wake (2009b); Wu et al. (2015); Zhang et al. (2008) |
| | Frequency Domain Reflectometry: frequency shift of reflected signal dependant on dielectric constant of soil | Ren and Vanapalli (2020); Romanovsky and Osterkamp (2000); Wu et al. (2021); Yoshikawa and Overduin (2005) |
| | Capacitance: resonant frequency of oscillating electromagnetic field across sample | Fen-Chong and Fabbri (2005); Fabbri et al. (2006); Lijith et al. (2021) |
| | Permittivity: measure of polarization of sample resulting from an applied electric field | He and Dyck (2013); Hoekstra (1966); Hu et al. (2020); Lara et al. (2020); Topp et al. (1980) |
| Specific heat | Differential Scanning Calorimetry: energy differential required to heat small sample | Kozlowski (2009) |
| | Adiabatic Calorimetry : energy absorbed in isolated chamber to result in heat change | Dillon and Andersland (1966); Stähli et al. (1999); Williams (1963); Watanabe and Wake (2009b) |
| | Isothermal Calorimetry: change in density of working fluid measured due to temperature change of sample | Anderson (1968); Anderson and Tice (1973); Jame and Norum (1973); Lovell Jr (1957) |
| | Heat Pulse: parallel probes emit heat and measure resulting soil temperature with thermocouple | Liu and C. Si (2011); Smerdon and Mendoza (2010) |





**Table A1.** Continued from previous page

| Property Measured | Method | References |
|---|---|---|
| Compressional wave speed | Ultrasound: speed depends on water/ice content | Timur (1968); Kolsky (1964); Nakano et al. (1972); yan Wang et al. (2006) |
| Absorption of gamma radiation | Gamma : determines total water (ice and liquid) in sample based on gamma absorption | Hoekstra (1966); Jame and Norum (1980); Zhou et al. (2014); Gray and Granger (1986) |
| Thermal conductivity | Thermal Conductivity: emitter and receiver of thermal signal over known distance | Wu et al. (2021) |
| Distance between particles | X-Ray Diffaction: distance between (only clay) particles based on diffraction of radiation | Morishige and Nobuoka (1997) |
| Equilibrium water content | Contact: dry sample placed in contact with ice and allowed to equilibrate at given temperatures | Chuvilin et al. (2008) |



**Table B1.** SFCC data template. Note that if data is reported in different units the original data is provided alongside the converted data.

| Index | Temperature [°C] | Volumetric Water Content [m³/m³] | Original Value [optional] |
|-------|------------------|----------------------------------|---------------------------|

**Table B2.** SFCC metadata template. Note that if data is not available, fields should be left empty.

| Metadata | Units |
|----------|-------|
| Index | NA |
| Author | NA |
| Year | NA |
| DOI | NA |
| Link | NA |
| Porosity | m³/m³ |
| Sand | kg/kg |
| Silt | kg/kg |
| Clay | kg/kg |
| Organic | m³/m³ |
| Density | kg/m³ (dry) |
| SSA | m²/g |
| SS/SLS | NA |
| Method | NA |
| Lab/Field | field/lab |
| Freezing/Thawing/Both | freeze-thaw/both |
| Saturation | m³/m³ |
| Reconstituted | 0/1 |
| Density Corrected | 0/[original units] |
| Depth | cm |

## Appendix B:  SFCC Data Template

Note that the index field in the SFCC data and metadata templates is a unique value assigned to each data series, and is used to link the data and the metadata. This field is added by the R package once the submitted data is accepted and added to the repository. In the metadata template (table B2), the fields should be interpreted as follows: The *Author* field indicates the first author of the publication, *Year* is the year of publication, *Link* is the DOI leading to the publication. Next follows the soil properties including soil *Porosity*, texture (*Sand, Silt, Clay*), and *Density*. These are reported either as mass or volume fractions

as is the practice when measuring them. *SSA* refers to the specific surface area of the sample, while *SS* or *SLS* indicates whether the soil grains interact directly with each other (SS) or include a liquid boundary layer between soil grains (SLS). This data is





frequently unavailable. Finally, there is data regarding the sample collection technique that indicates if the data is collected in the *Lab* or in the *Field*, which limb (*Freezing/Thawing/Both*) of the SFCC is measured, the degree of saturation of the sample, and whether the sample is undisturbed or *Reconstituted* from a dried, packed, soil sample. Finally there is a flag (*Density*

*Corrected*) to indicate if the measurement units were different from $m^3/m^3$ and if unit conversion was necessary. If the units were converted, then the original units are included in this field, and if the data are reported in $m^3/m^3$, the value is 0. If the measurement was made in the field, or if there is data regarding the soil sampling *Depth*, this is also included.



*Author contributions.* Élise Devoie gathered data, constructed repository, developed R package, drafted manuscript. Stephan Gruber and Jeffrey McKenzie provided recommendations, revised manuscript, and assisted in structuring data

*Competing interests.* The authors declare that they have no conflict of interest.

*Acknowledgements.* The authors would like to acknowledge Nick Borwn for significant contributions to the R package development, Dr. Boike and Dr. Flerchinger for providing data, as well as Dr. Sherepenko for assistance in data digitizing.





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
