# Peer review of "A Repository of Measured Soil Freezing Characteristic Curves: 1921 to 2021"

_Earth System Science Data, 2022_

## Author Response (AR1)

*Thank you very much for your careful review and excellent suggestions. The authors appreciate the feedback and intend to make the changes noted below to the manuscript before publication.*

Major

The critical point on data uncertainty is left to an accompanying paper, Devoie et al., 2022b, in prep., which is not available at this point? As a potential user of the database, having some quality assessment of the different data sets in the data base (or even data uncertainty in the best of worlds) is critical. As an example, if one wants to look into the freezing point depression and a measurement at -0.09 degree C is reported, it is important to know if the temperature sensor has an accuracy of 0.01 K or 0.1 K. I assume that such information is extremely hard to come by in most cases, but outsourcing this uncertainty part to an external manuscript (which is not even available yet) is not a good solution. So ideally, the authors should include uncertainties or quality assessments in the database wherever possible, and discuss this briefly in the manuscript. If such data quality assessment cannot be provided for any of the data, this should at least be mentioned, as it is a highly important limitation for some applications. On a general note, the ESSD paper should contain all aspects related to data use, and uncertainty is a rather important aspect. This still offers the possibility for a companion paper on the measurement techniques and related uncertainties.

*Response*
*The authors agree that the dataset is of limited value without the error estimate, but the error is rarely reported in the initial publication. Appropriate error estimation and  is so involved it requires more supporting information than is appropriate for an ESSD paper. The archive is however flexible, and zenodo allows to add data. Once the second paper is complete (and peer reviewed), the authors will publish uncertainty estimates alongside each measurement. This will be noted as a limitation of the current dataset.*

The csv file for metadata contains doi's and author + year, but not the full reference. Are all published studies also referenced in the main paper, so that the citation can be found there? If yes, it would be good to mention this somewhere, so that the authors of the original studies can receive due credit if their data are downloaded from the database and used. It could be mentioned e.g. under "data availability" that you encourage to additionally refer to the original publication whenever possible. If no, it would be good to provide a list of the references of all published studies somewhere so that it can be linked to the data in the database. Potentially related to this, in SFCCMetadata.csv I am not sure about the last column "file". Are these pdf's available as a collection somewhere, or is this only an internal file path? In case of the latter, consider replacing it with the full citation!?

*Response*
*Yes, the references are included in the main paper, and the authors appreciate the suggestion to refer to the original data and will add that to the data availability statement. The R package included constructs the reference from the DOI of the publication, but the authors will add a metadata field for the full reference. Updated data availability statement:*

"The SFCC data in support of this publication are openly available as a zenodo archive: https://doi.org/10.5281/zenodo.5592825 (Devoie et al., 2022a), which will be regularly updated as users contribute to the database. Users are encouraged to cite the original source of the data they extract from the repository, and a citation is provided with each dataset. The R package, including this data and data tools, is available from the author's github: https://github.com/egdevoie/SFCCRepository (Devoie and Brown, 2022), and work is underway to publish this as a CRAN package."

Add a section on how new entries to the database will be quality-controlled in the future. Otherwise, measurements of questionable quality may be added in the future and weaken the usability of the database. See Minor comments.

*Response*
*Excellent suggestion. After discussion, the corresponding author has decided to curate the database personally, so any additional measurements will be evaluated before being added to the repository. One of the tools in the R package performs the assessment of data quality and will be used to ensure that the metadata is as complete as possible, and the data fall within the expected ranges. This will also help to ensure that the zenodo archive and the data attached to the R package are identical. The invitation to contribute to the database will be modified to reflect this:*

"5.1 Add an SFCC to the repository

The first data tool is a function which takes SFCC data alongside its associated metadata and adds it to a local copy of the existing repository. This file contains the SFCC data and is linked to an index in the metadata file. The metadata file stores the information associated with the SFCC. To be appropriately interpreted, the SFCC data to be ingested must be prepared according to the data template in Appendix B. The script first verifies that all required data is present and that the data to ingest does not already exist in the archive before adding it. Users are encouraged to contact the authors to contribute their datasets to the larger repository, making them available to the wider cryosphere community. This will also ensure the accompanying data tools can be applied to the dataset. The zenodo archive will be updated by the authors with new user contributions after detailed assessment of the quality of the data and accuracy of the metadata."

l.184: I can't access https://github.com/egdevoie/SFCCRepository , this gives an error for me. It may be a problem on my end, but worth checking.

*Response*
*The github repository is not yet published as an official R package on CRAN (this will happen simultaneously with the second publication in this series) and so may not be accessible to all users? If the issue persists the authors would be happy to provide access or upload the individual components of the package though it is not ready for distribution to R users in general.*

Minor
I am unsure about the title. Is the fact that 100 year old measurements are included really that important? As a user, I would mostly care about having the best possible measurements, no

matter how old they are. A measurement conducted 100 years ago should have yielded the same result for the same soil as a measurement conducted today, so unlike quantities that do change over time (e.g. climate-related data), the "length of the time series" does not play a role here. But I leave this point up to the preference of the authors.

*Response*

*Proposed new title:* "A Repository of Measured Soil Freezing Characteristic Curves: 1921 to 2021"

l. 3: very minor point: "the soil ice content" is also determined by the water content before onset of freezing (i.e. water plus ice content). So something like "partitioning of soil water and ice" would be more precise.

*Response*

*Thank you for the suggested clarification – agreed.*

l. 54: it would be good to add a sentence on dissolved ions and other molecules in the soil water which depress the freezing point and can concentrate in the brine upon freezing, leading to further depression of the freezing point of the brine.

*Response*

*Also agreed, this archive does not include the effects of salinity, though some measurements have been made, and the authors hope to explore that aspect of the data in the future. This sentence will be added with reference to the prior work on salinity in SFCCs:*

"The freezing point depression is further complicated by the presence of solutes which further depress the freezing point and can play a dominant role in determining the residual water content and shape of the SFCC in saline soils (Amankwa et al. 2021). A detailed study of solute effects is left for future work, and the data presented herein is focused on non-saline systems."

l.61: add year in reference

*Response – yes, thank you.*

l.150: I do not agree with the causality. If the last water to freeze is also the last water in drying soils, this only means that the low-temperature end of the SFCC can be approximated by looking at drying soils. For the rest of the SFCC range, this is not necessarily true, and it is in fact contested by some studies (e.g. overview in Karra and Painter, 2014).

*Response*

*The authors were perhaps unclear – this causality only applies at low temperatures/residual water contents. The sentence has been re-written as follows:*

"This assumption means that saturated SFCCs can be used as an adequate approximation of unsaturated SFCCs at temperatures where the residual unfrozen water content is less than the water content of the unsaturated soil. The same relation between unfrozen water content, pressure (saturation) and temperature can therefore be used for both saturated and unsaturated soils at cold temperatures (Dall'Amico et al., 2011)."

l.170: are you sure about 1, and not rather "can exceed the porosity of the dry soil"?

*Response*

*Well, both actually, but the authors agree that 'can exceed the porosity of the dry soil' is more clear. Updated in text, thank you.*

Appendix B: it would be good to add a field on "known errors/uncertainties", e.g. if authors are not sure whether the position of the temperature and soil water sensors may have shifted during measurements. This could be essential to quality-control new entries to the database, see Major comments.

*Response*

*Yes – more to come in publication 2, the authors completely agree.*

**Anonymous Referee #2**

*Thank you very much for your careful review and suggestions. The authors appreciate the feedback and intend to make the changes noted below to the manuscript before publication.*

Comments on "A repository of 100+ years of measured soil freezing characteristic curves" by Devoie et al. The authors described a repository for some measured soil freezing characteristic data, which are helpful for the community. The current manuscript is too coarse for publication, a lot of information is missing, and the logic is unclear, although it is a data paper. I would like the have the authors add more details to make a complete paper before considering it for publication in ESSD. The introduction is short, why the SFCC is important for the community. A short paragraph with a soft touch on the soil freeze-thaw model is insufficient. Some background here for the SFCC and its implication for the community is valuable, and the knowledge gaps (or data gaps) should be provided.

*Response*

*The following paragraphs will be added to the introduction:*

"The first SFCC in literature was measured in 1921 using the difference in volume between water and ice to measure the ice content of samples below the freezing point (Bouyoucos 1921). Since this first publication, SFCCs have been measured using various techniques in lab and field settings. Koopmans (1966), Spaans (1995), Bitelli (2004) and Azmatch (2012) have extended SFCC data to extrapolate the relationship between water content and matric potential (pressure). This method has the benefit of extending the soil moisture curves to very negative pressures but does have some limitations presented by Ren (2019). These include the assumption that the liquid and solid phase are in thermodynamic equilibrium, limiting the freeze/thaw rates to be very slow (Liu 2011). The near-zero temperatures where much of the phase change occurs at pressures near the air entry pressure are difficult to accurately resolve, and there are mechanistic differences in the processes; including the diffusion of gas to form air bubbles in drying which has no symmetric behaviour in freezing, and ice nucleation which has no symmetric behaviour in drying. Solutes further complicate this (Ren 2019).

Much of the SFCC data have been collected in the lab setting on fine-textured soils, though it is known that the sample history and number of freeze-thaw cycles affects the measured curve (Buehrer 1943, Kozlowski 2004). There is little data concerning coarse sands or gravels as they are challenging to measure, and there is little comparison between data collected on similar soils in different settings or using different techniques. The studies that have been performed show that sample preparation is critical in defining outcomes (Kozlowski 2004), the choice of measurement method may impact outcomes depending on soil texture (Smith 1988, Yoshikawa 2005), and there are strong hysteretic effects between freezing and thawing, especially in fine-grained soils (Tian 2014, Lara 2020). Many other patterns and comparisons may be made concerning soil texture, uncertainty and error in measurement, partially saturated soils, and other factors, but have yet to be possible due to a lack of ability to compare the wealth of measured data in literature. This repository has as its aim not only to support modelling exercises with

data-driven SFCC estimates, but also to construct more robust comparisons between SFCCs and improve our understanding of freeze-thaw processes in soils."

The background section does include an example of SFCC, a typical SFCC curve. I suggest it should be included in the dataset section. It can be used as a case study to show what has been done in the current repository. If the authors would like to keep it as it is as a background for the SFCC, I would suggest the authors put out some data from your current repository and compare them with this figure to show better present the data and confirm the SFCC introduced in the early section of the paper.

 *Response*

*The figure in the background section does not have numeric axis labels and is intended as a schematic of an SFCC, with important values labeled using their parameter names. The authors are hesitant to include data in this figure as it would detract from its versatility, and a choice would need to be made on the model used to describe the data and fit the curve. This leads to questions about uncertainty and model choice etc. The authors will instead included an additional figure: an example of the use of the R package to select data based on soil texture (for sandy soils) – Figure 4.*

If the data contain lab/long for data points. I would say the coverage of the data points is missing. Although some data are extracted from lab experiments, as the soil type is available, it is possible to put those data on a global map for a geographic location.

 *Response*

*The authors are unsure of exactly what this comment is recommending, however it is unfortunately not possible to create a map of where the tested soils were collected as this information was not presented in many of the publications. For example, textural data and information is provided, alongside a soil name such as 'Norway Clay' or 'Bentonite' (Smith & Tice 1988), which is insufficient to meaningfully place the data in spatial context. A vast majority of the data is collected in lab settings however a field will be included in the metadata for location of field samples going forward.*

The size of the dataset, the primary format of the data, storage, and accessibility are missing; if possible, I would like to see a few robust patterns based on the data, but all of them are missing.

 *Response*

*Much of this information is included in section 3 : Data Aggregation. The following lines have been modified:*

"…The digitized curves along with their metadata are stored  in .csv format in an open-source archive for future use, and an R package containing the data and data tools is discussed in section 5. The repository currently contains 30 MB of data, corresponding to approximately 870000 data points."

*The authors feel that identifying robust patterns in the data is beyond the scope of the data publication but anticipate further work with the dataset to explore these.*

It is good to see the R package is included in the repository; the authors should provide more detailed information about the R scripts, the essential functions, and the organization of those functions. Some examples of the analysis should be provided.

*The corresponding author intends to submit a publication to JOSS, as well as publish the package on CRAN. This level of documentation will be more specific to users of the R package and the authors feel that including more detail in this data publication diverts the focus from the data presented to the data-handling tools available to manage it.*

Some writing problems stand out and should be addressed. I just listed a few; authors should be more serious about this issue.

*The authors have proof-read the updated version of the manuscript and carefully corrected the typographic errors noted below.*

Line 23, an example of SFCC – *disagree "an example soil freezing characteristic curve" is not a problem.*

The 100+ years is not appropriate, as you compiled data of >100 years apart; I originally thought you had data for 100 years duration.

*Title revised to* "A Repository of Measured Soil Freezing Characteristic Curves: 1921 to 2021"

Line 155, this package contains the repository. I thought you mentioned your repository contains the R package, but you stated the R packages contain the repository. I guess you meant your R package contains the function to call for the data repository and can directly access the dataset.

*The R package is linked in the description of the repository, and a copy of the repository is found within the R package such that users might add data to a local copy of the repository for specific use.*